# Development of a Modular Tissue Phantom for Evaluating Vascular Access Devices

**DOI:** 10.3390/bioengineering9070319

**Published:** 2022-07-15

**Authors:** Emily N. Boice, David Berard, Jose M. Gonzalez, Sofia I. Hernandez Torres, Zechariah J. Knowlton, Guy Avital, Eric J. Snider

**Affiliations:** 1U.S. Army Institute of Surgical Research, JBSA Fort Sam Houston, San Antonio, TX 78234, USA; emily.n.boice.ctr@mail.mil (E.N.B.); david.m.berard3.ctr@mail.mil (D.B.); jose.m.gonzalez355.ctr@mail.mil (J.M.G.); sofia.i.hernandeztorres.ctr@mail.mil (S.I.H.T.); zechariah.j.knowltown.ctr@mail.mil (Z.J.K.); guy.avital.md.il@gmail.com (G.A.); 2Trauma & Combat Medicine Branch, Surgeon General’s Headquarters, Israel Defense Forces, Ramat-Gan 52620, Israel; 3Division of Anesthesia, Intensive Care & Pain Management, Tel-Aviv Sourasky Medical Center, Tel-Aviv 64239, Israel

**Keywords:** model development, automation, tissue phantom, vascular access device, medical devices, porcine, human, femoral, hypovolemia, nerve fiber

## Abstract

Central vascular access (CVA) may be critical for trauma care and stabilizing the casualty. However, it requires skilled personnel, often unavailable during remote medical situations and combat casualty care scenarios. Automated CVA medical devices have the potential to make life-saving therapeutics available in these resource-limited scenarios, but they must be properly designed. Unfortunately, currently available tissue phantoms are inadequate for this use, resulting in delayed product development. Here, we present a tissue phantom that is modular in design, allowing for adjustable flow rate, circulating fluid pressure, vessel diameter, and vessel positions. The phantom consists of a gelatin cast using a 3D-printed mold with inserts representing vessels and bone locations. These removable inserts allow for tubing insertion which can mimic normal and hypovolemic flow, as well as pressure and vessel diameters. Trauma to the vessel wall is assessed using quantification of leak rates from the tubing after removal from the model. Lastly, the phantom can be adjusted to swine or human anatomy, including modeling the entire neurovascular bundle. Overall, this model can better recreate severe hypovolemic trauma cases and subject variability than commercial CVA trainers and may potentially accelerate automated CVA device development.

## 1. Introduction

Endovascular resuscitation is a rapidly evolving field for managing severe traumatic hemorrhage, nontraumatic sources of acute blood loss, and acute medical illness characterized by hemodynamic decompensation [1]. These injury types can lead to cardiac arrest and death if not addressed quickly, as hemorrhage remains the leading cause of preventable death for civilian and military causalities [2,3]. The first critical step for treatment is gaining central vascular access (CVA), meaning access to a central vein or artery, which allows for interventions including hemorrhage control [4,5], extracorporeal perfusion support [6,7], fluid infusion [8,9], as well as drug delivery [10].

CVA, when performed quickly and successfully, can be lifesaving. But emergency usage remains challenging due to several factors. Even using ultrasound guidance, it is oftentimes difficult to locate vessels due to loss of arterial pulse and vasoconstriction during hemorrhage [11]. Edema [12] and anatomical complexities can also obscure visualization. In pre-hospital settings with limited lighting and evacuation opportunities, this technique becomes even more challenging. The femoral route is often preferable due to the lower risk of early, mechanical complications [13].

Automated medical devices that rely on artificial intelligence algorithms to assess ultrasound images or videos and guide proper needle insertion to facilitate CVA are currently in development. These devices direct the end user to identify the correct location and either use a needle guide for manual insertion or robotics to deploy a needle as a first step for insertion of catheters using the Seldinger technique. Once catheter placement is confirmed, lifesaving therapeutics can be administered through vascular access. CVA automation has been shown to require little training to use the device while still maintaining high performance accuracy for needle placement [14].

As more CVA medical devices are pushed forward to market, there are limited ways to test and objectively measure performance, which is essential for device development and troubleshooting, as well as to identify CVA devices most suitable for healthcare providers [15]. Performance of such a device will only be as good as the testing platform utilized for its development.

### Related Work

A number of ultrasound compliant phantoms are commercially available and could serve to assess automated vascular access devices. These include the Gen II Femoral Vascular Access & Regional Anesthesia Ultrasound Training Model (CAE Healthcare, Sarasota, FL, USA), FemoraLineMan System (Simulab, Seattle, WA, USA), and the Vascular Access Training Phantom (CIRS, Norfolk, VA, USA). Phantoms range from simple flat testbeds to more complex forms resembling the human lower torso and upper limbs. These phantoms, while straightforward for medical training, have limitations for use in testing automated CVA devices. These trainer systems offer simplistic, non-realistic US images, without the ability to test in various locations across the body and in differing volemic states [16]. Incorporating modifications to these test platforms would allow the end user, or the automated device, to detect more multifaceted situations, such as hypovolemic states, the presence of complex anatomy, or the use of doppler for confirmation of vessel identity.

Live animal models can also be utilized for testing CVA devices. Typically, this requires the use of a large animal, such as a porcine subject, and is a costly experiment due to the utilization of medical facilities, trained vivarium personnel, and anesthesia of the animal [16]. Recent work has developed an ex vivo pseudo-perfused porcine lower body model allowing for high-fidelity sonographic imaging and simulation of normovolemic and hypovolemic conditions [17]. As most labs have access to abattoirs, this platform can be readily available for testing CVA devices. However, its setup is cumbersome and requires training. A simple, cheap phantom model can aid in the initial testing phases, potentially reducing the number of animal experiments required, and thus animal suffering and costs.

Here, we present a simple benchtop platform utilizing a tissue phantom model produced of gelatin and cast in a 3D printed mold with channels for vessels. This model is modular in design, allowing for different vessel sizes, locations and incorporation of bundles mimicking nerve fibers. This phantom can be embedded with latex tubing for vessels and integrated into a flow loop with pulsatile fluid flow and circulating fluid pressure control. These vessels can be removed and evaluated to quantify damage to the vessel wall. The ultimate goal of this work is to develop a standardized platform for assessing and comparing automated CVA devices.

## 2. Materials and Methods

### 2.1. Custom Phantom Mold Construction

The overall mold design was based on measurements of a human femoral region. The cylindrical phantom was 120 mm in diameter and 106 mm long. Components of a three-piece mold cavity were 3D modeled using computer aided design (CAD) software (Autodesk Inventor, San Rafael, CA, USA), and fabricated using a fused deposition modeling 3D printer (Raise3D, Irvine, CA, USA) with Polylactic Acid (PLA, Raise3D, Irvine, CA, USA). The inner surface of the mold cavity was coated with a resin (XTC-3D, Smooth-On, Easton, PA, USA) to prevent leaking and produce a smooth surface finish on the mold. The two halves of the mold cavity snapped together using a key slot-style seal and were capped with a lid. Blind and through holes were included in the floor and lid of the cavity, respectively, to support a nylon “bone” and removable inserts for the channels used for the vessels.

The bone had a diameter of 25 mm, the average diameter of a femur [18,19]. Arteries and veins were designed to simulate vessels in normovolemic and hypovolemic states based on measurements taking from swine tissue and were used as inserts in two regions of the model (Figure 1A). The vessels were also designed using CAD software but were fabricated using a higher resolution stereolithographic Form 3 printer and Durable resin (Formlabs, Somerville, MA, USA). For the porcine mold, the mean diameters of the normovolemic artery and vein were 4.2 mm and 4.9 mm, respectively, while the hypovolemic artery and vein were 4.2 mm and 2.8 mm, respectively. The vessels began lateral to one another at the superior end of the mold, 10.5 mm below the surface. Then, the vein gradually wrapped around the artery from a lateral to a posterior position at the inferior end and dove to 18 mm below the surface, while the artery remained at the same depth. For convenience, STL files for the various phantom designs are included as Appendix A.

#### Modification for Human Anatomy and Nerve Fiber Inclusion

For human anatomy, the same outer tissue mold was used but with a different lid and floor design as well as changes to the vein and artery sizes. The normovolemic and hypovolemic arteries were 7.4 mm and 5.8 mm in diameter, respectively. The veins had 7.4 mm and 4.2 mm mean diameters for normovolemic and hypovolemic states, respectively [14,20]. The relative orientations of the artery and vein were the same as the porcine model with the artery positioned at a constant depth of 18.4 mm beneath the tissue surface. The vein began at 22.4 mm beneath the surface at the superior end and adjusted to 25.1 mm at the inferior end of the mold.

Next, we introduced an additional channel for a nerve fiber sheath to the human-mimicking phantom (Figure 1B). A basic configuration resulted in a separate triangular shaped channel (Figure 1B, brown channel) where the nerve fiber was 4.8 mm in height and 5.4 mm in width and positioned 1.1 mm from the artery and 11.3 mm from the vein. A more advanced use case was constructed where the three channels combined into a single neurovascular bundle within the mold (Figure 1B, purple teardrop channel). A surface area of 59.8 cm^2^ was used for the vessel size which allowed for inclusion of all three features, with the vein accepting a more realistic oval shape. The nerve fiber was twisted 90 degrees along the length of the model to add more variability to its design (Figure 1B).

### 2.2. Gelatin Tissue Phantom Preparation

Phantom was constructed using 15% (*w*/*v*) gelatin (Thermo-Fisher, Waltham, MA, USA) and 0.5% (*w*/*v*) flour dissolved in a 2:1 solution of water and evaporated milk (*v*/*v*) (Nestle, Vevey, Switzerland). When the 3D printed molds were constructed and vessel inserts added, a pair of clamps were used to secure the halves together and tape was applied to the seam to improve leak resistance. The gelatin solution was poured into the mold and refrigerated at 4 °C until the phantom solidified, approximately 90 min. When solid, the gelatin phantom was removed from the mold and the vessel inserts were carefully removed to leave empty artery and vein channels.

#### 2.2.1. Tubing Addition to the Phantom Mold

Next, the porcine phantoms were plumbed with 175 mm lengths of 1/32” (0.079 cm) wall thickness latex tubing (Cole Parmer, Vernon Hills, IL, USA) of varying diameters based on the state of the artery or vein (Figure 1C). To minimize trapped air in the vessel channels, the gelatin phantom was submerged under water while the latex tubing was inserted. Tubing was inserted into each vessel channel (see Figure 1C for list of tubing sizes for each use case). The phantom was then attached to the flow loop platform (Section 2.3).

#### 2.2.2. Various Modifications to the Gelatin Phantom

For the human phantom model, the same methodology was followed except quarter inch diameter latex Penrose drains (Graham-Field Health Products, Atlanta, GA, USA) were used instead of bulk latex tubing. For the nerve fiber model, a 15% (*w*/*v*) gelatin nerve fiber solution was created with 1% (*w*/*v*) flour in 2:1 (*v*/*v*) water and evaporated milk. This solution was poured into the nerve fiber slot after removing the insert. A silicone rubber sheet (McMaster-Carr, Elmhurst, IL, USA) was placed under the phantom during pouring to minimize fluid leaking as it solidified in the nerve fiber slot. For the neurovascular phantom, normovolemic human artery/vein tubing was held vertically in place by a ring stand. The space around the vessels was filled with the nerve fiber bundle solution and allowed to solidify.

Lastly, a skin layer was fabricated by using EcoFlex 50 mixed with 0.25% (*w*/*v*) flour (Smooth-On, Easton, PA, USA). After de-gassing to remove air bubbles, the silicone solution was poured into a rectangular container to form a 2 to 4 mm thick layer. After polymerization of the silicone elastomers, the skin sheet was draped around the phantom during use, making sure to not trap air between the skin and phantom surface.

### 2.3. Integration of the Gelatin Phantom into a Benchtop Flow Loop Platform

A SuperPump (ViVitro Labs, Victoria, BC, Canada) was used for generating pulsatile, cardiac mimicking arterial flow within the flow loop test platform (Figure 1D). Three different waveforms provided by the manufacturer were used for creating different waveforms with the tissue phantom. The stroke volume on the pump was adjusted to achieve desired systolic and diastolic pressure magnitudes. From the cardiovascular pump, silicone tubing passed through a three-way valve where arterial pressure was recorded by a pressure transducer (ICU Medical, San Clemente, CA, USA), upstream of the tissue phantom (Figure 1D). Waveforms were visualized by patient monitor (Draeger, Lϋbeck, Germany) connected to pressure transducers (ICU Medical, San Clemente, CA, USA) on the arterial and venous side and recorded using PowerLabs data acquisition hardware (AD Instruments, Sydney, Australia). Next, the tubing line from the cardiovascular pump was connected by barb fitting to the arterial tubing line in the tissue phantom and the downstream side was connected to an additional section of silicone tubing connected to the cardiac pump return. This created a closed system for the arterial side in which air bubbles were removed by three-way valve attached to the pump-head. The total volume of the closed system was adjusted by syringe to raise or lower the overall MAP of the system. For closed loop arterial flow experiments, tap water was used as the circulating fluid on the arterial side.

The venous tubing in the phantom was connected to a hydrostatic reservoir and pressure transducer upstream of the phantom. The other side of the venous tubing was closed off after air bubbles were allowed to leave the tubing line (Figure 1D). The height of the hydrostatic reservoir was adjusted to keep the venous pressures within physiological normovolemic and hypovolemic ranges. Water was used as the fluid for the venous side of the flow loop.

#### Flow Loop Modifications for Doppler Compliance

When showing the doppler capabilities of the tissue phantom, flour was added at 2 g/L to the circulating water. Due to the insolubility and density discrepancy between flour and water, a reservoir and magnetic stir plate (Thermo-Fisher, Waltham, MA, USA) were added to the loop between the phantom and the peristaltic pump. This ensured the flour remained in solution for arterial flow in the phantom. However, this opened the system so that pressure could not be increased by adjusting the system volume. Instead, a C-clamp (McMaster-Carr, Elmhurst, IL, USA) was added downstream of the gelatin phantom in the arterial loop to increase resistance in the flow loop, simulating adjustable arteriolar resistance.

### 2.4. Pressure Integrity of Punctured Vessels

The integrity of the punctured vessels was evaluated by a leak rate assessment [21,22,23,24] of the 1/8 inch OD latex tubing (Figure 2). Samples of the arterial tubing were cut to standardized lengths of 175 mm (*n* = 4) and placed in line with a pressurized bag of solution, three-way valve, and a pressure transducer (SP844, MEMSCap, Isere, France), which was connected to a PowerLabs Data Acquisition unit (AD Instruments, Sydney, Australia). To reduce the effects of surface tension on pressure measurements, dish soap (Concentrated Dishwashing Liquid, Procter & Gamble, Cincinnati, OH, USA) was added to the water in the system at a 0.0125% (*w*/*v*) concentration. A starting pressure of approximately 120 mmHg was applied using the pressurized bag, and the system was allowed to equilibrate. The segment of tubing was then isolated from the pressurized bag but remained connected to the pressure transducer. The pressure-drop over time (2 min) was recorded with LabChart software (ADInstruments, Sydney, Australia) to evaluate a correlation between the number of punctures and leak rate. Prior to introducing tubing punctures, a control test was conducted to confirm the system base leak rate was minimal. Using an 18 G spinal tap needle (Sensi-Touch, Sherwood Medical, St. Louis, MO, USA), holes were introduced 1 cm apart sequentially and recorded to quantify the effect each additional tubing puncture had on leak rate. Data were averaged and analyzed using MATLAB (MathWorks, Natick, MA, USA). From the initial pressure of 120 mmHg, the pressure after 5 s was tracked for comparison across all tubing and puncture amounts (*n* = 4 tubing with 0 to 5 punctures).

### 2.5. Ultrasound Imaging

Ultrasound imaging was performed using the Sonosite Edge II Ultrasound system (Fujifilm, Bothell, WA, USA) with a HF50 transducer (Fujifilm, Bothell, WA, USA). B-mode images and video clips were obtained with both in plane (IP) and out of plane (OOP) probe placement at varying depths and gain settings, to achieve a clear view of the vessel configurations. For baseline image acquisition, the phantom was submerged in water during imaging to eliminate the air interface. With the flow loop modification capable of acquiring doppler waves, the ultrasound probe was positioned OOP along the target arterial vessel with fluid flow. The Sonosite Edge system was also used to collect pulsed wave doppler waveforms by aligning the doppler gate to the center of the target vessel to acquire signal and sound.

### 2.6. Ultrasound-Guided Vascular Access

Aquasonic Clear Ultrasound Gel (Parker Laboratories, Inc., Fairfield, NJ, USA) was applied to the surface of the gelatin tissue phantom and ultrasound probe was guided along the surface to observe OOP view of the vessels. As the target vessel was identified and brought into the center of the image, an OOP ultrasound image of the vessels was obtained. Using a 2.5 in, 18-gauge needle (Sensi-Touch, Sherwood Medical, St. Louis, MO, USA) inserted with an out-of-plane (OOP) approach at an approximate angle of 45° to the surface, real-time imaging visualized the needle advance through the phantom into the targeted vessel. Successful cannulation was confirmed by aspiration of fluid into the syringe. Following insertion, confirmation of needle placement was documented using in-plane (IP) images.

## 3. Results

### 3.1. Phantom Waveform Results

After the molds were fabricated and the gelatin phantom was created for mimicking swine anatomy (Figure 1A), latex tubing was integrated within vessel channels so that the phantom could be connected to a flow loop platform for mimicking physiological flow (Figure 1D). Three physiological waveforms for the cardiovascular pump were setup in both normovolemic and hypovolemic conditions (Figure 3A–C). A summary of systolic, diastolic, and mean pressures is shown in Table 1. Lastly, we evaluated the capability of the phantom platform for doppler flow using flour as a heterogeneous reflecting agent during fluid flow. Overall, using this methodology, doppler signal and waveform signal propagation were evident for the phantom (Figure 3D).

### 3.2. Swine Phantom Ultrasound Results

Next, we evaluated the swine phantom capability for ultrasound imaging. Latex tubing of various diameters was used to mimic veins and arteries within the tissue phantom and an external silicone layer was added to mimic skin. Overall, the gelatin phantom was ultrasound compliant, and the flour contained within the phantom bulk, as well as the evaporated milk fat, provided a hyperechoic ultrasound signal beyond the traditionally anechoic properties of gelatin hydrogels (Figure 4A). Needle insertion into the artery and vein was tracked in normal physiological conditions out of plane (Figure 4A artery, Figure 4B vein). As a result of the latex tubing used in the phantom, vessels were too rigid to collapse when mimicking hypovolemic levels just by reducing pressure in the system alone. Instead, smaller vessel sizes were integrated into the phantom with smaller latex tubing to mimic the reduced size of the artery and vein. Needle insertion could be tracked within hypovolemic levels for out of plane (Figure 4C artery) and in plane views (Figure 4D vein).

### 3.3. Pressure Integrity of Punctured Vessels

Another advantage of latex tubing within the tissue phantom was it can be removed and evaluated for damage due to the needle insertion procedure. A methodology was developed to evaluate how well the tubing resists leaking by measuring pressure decay versus time in a fluid system (see Section 2.4 Pressure Integrity of Punctured Vessels). For proof of concept, tubing pieces were evaluated with 0 to 5 holes sequentially punctured with an 18 G needle spaced 1 cm apart. The effect of the number of needle insertions on the pressure decay rate was evident (Figure 5A), with the decay rate increasing with each hole. Based on the rates of decay, a five second interval was set where optimal differences in performance were evident (Figure 5A—shaded region). Evaluating four different tubing pieces resulted in the dynamic operating range for this test method being between one and three holes in the tubing as more holes did not consistently increase the pressure decay rate (Figure 5B).

### 3.4. Human Mimicking Phantom Results

Finally, we highlighted more of the phantom’s capabilities by introducing additional features to the model. First, the model’s vessel sizes and depths were modified to human anatomy. Again, using the silicone skin layer, latex tubing for vessels, and flow loop platform, a normal (Figure 6A) and hypovolemic (Figure 6B) model were developed and used successfully for manual vascular access. While similar to the swine image sets, the vessel locations and sizes for high and low pressure were modeled after published values [14]. Next, we integrated a nerve fiber bundle to increase the complexity and potential use cases beyond vascular access. The nerve fiber bundle was comprised of a higher hyperechoic property than the bulk. We first spaced the nerve fiber bundle and the vessels out similarly (Figure 6C). The nerve fiber bundle is evident and can be distinguished from the bulk phantom tissue properties as well as the vessels. However, the small size of the nerve fiber region looked hypoechoic instead of the anticipated hyperechoic, likely due to the gelatin not completely filling the channel before solidification. Next, we evaluated using a single channel cutout for a neurovascular bundle comprised of artery, vein, and nerve fiber (Figure 6C). This methodology was successful for introducing the nerve fiber and for allowing the vessels to be at closer proximity than the other phantoms shown.

## 4. Discussion

Central vascular access (CVA) is a valuable medical procedure for life-saving interventions and invasive monitoring in many severe trauma situations. To reduce the cognitive load on medical personnel, automated CVA medical devices are being developed that will simplify emergency medicine and provide access to life-saving procedures not normally possible in combat casualty care. Medical device development can often be stifled by inadequate testing platforms which may be the case for CVA, as key features such as physiologically relevant flow and modeling both normal and hypovolemic vessel sizes is not possible with current methods. We believe the tissue phantom model developed here fills this key capability gap and can accelerate medical device development.

The phantom platform introduced in this work is first and foremost flexible to the end user’s need, allowing it to provide several features that may be essential during device evaluation. Many of the flexible design features can be accommodated by modifying CAD files prior to 3D printing and phantom construction. The vessels can be positioned deeper in the phantom or more superficial for mimicking different central vessels in the body. Further, the proximity of the artery and vein can be set to as close as required. This is only possible with this phantom platform, as latex tubing is used for the vessels. Previous iterations of phantom development used empty channels and barbed fittings for incorporation into the flow loop. The flow rates required for physiological pressures compromised the phantom integrity between the vessels. For further complexity, we show that the phantom can be fabricated with a nerve-simulating component, creating a more realistic neurovascular bundle. Alternatively, this phantom design may be suitable for evaluating regional nerve block technique and medical devices as they become available. Another design feature that can be captured with the modular phantom design is the underlying species physiology being modeled. We highlight swine and human use cases, where positioning and sizes were mimicking medical imaging captured for both, but this can be extended to any number of species. Swine anatomy was added to the tissue phantom as it could potentially aid bridging the transition gap of product testing from human phantom models to live swine testing as the vessels are different sizes and at different depths. Paired with modifying the overall size and shape of the phantom, this can be further extended into modeling vascular access for anatomies outside of the femoral region.

Another key design feature for the developed phantom platform was maintaining vessel access on both ends of the phantom to allow for continuous fluid flow. Many tissue phantoms utilize a hand-pump and a closed off vasculature, which makes mimicking a cardiac waveform challenging and hypovolemic conditions impossible. With the flow through design, the phantom platform was attached to a flow loop containing pressure monitoring equipment and a heart-mimicking pump. Realistic waveforms were recorded with the system and, further, the inclusion of an ultrasound scattering agent such as flour to the circulatory lines allowed for an ultrasound doppler signal. This may be critical for medical device designs using a doppler signal as a means of distinguishing arteries and veins. With full hydrodynamic control of the test platform, normal and hypovolemic scenarios could be configured with the test platform. Pairing this with adjusting the tubing sizes during fabrication, allowed for fully replicating the hypovolemic condition in a tissue phantom, as this can be one of the largest challenges for vascular access, especially venous access, during severe trauma situations. Lastly, with the channels being embedded with tubing, this tubing can be removed post-puncture to allow evaluation of leak rate from the tubing, providing a more quantitative metric for evaluating the needle-inflicted trauma to the vessel wall. A summary table of features of the designed phantom and its comparison to conventional commercial phantom is shown in Table 2.

However, there are some limitations with the phantom platform developed. First, the gelatin medium used is homogeneous, which may not properly mimic tissue level organization found in muscles, for instance. This may not challenge automation features used in certain CVA devices, however, this can be resolved by the introduction of ultrasound complexities in the tissue phantom, as we have previously described [25,26,27]. Second, the surface geometry of the phantom is simplistic and does not contain the inguinal crease which is often used for aligning vascular access attempts. For fabrication simplicity, this was not accounted for in its current design while other design aspects were configured. This feature can be accommodated by placing the phantom in a larger anatomical model or modifying the surface design of the current model, together with a modification to the bone simulating component, to meet this use case. This limitation also prevents effective use of this model for training and skill poling purposes, as it does not incorporate some key important challenge elements relevant for the human performer, mainly identification of the relevant surface anatomy. The purpose of this model is to assist in the development and evaluation of automated CVA devices. Such a specialized application requires customization of the model to the requirements that cannot be fulfilled by models designed for human performers training.

Next steps for this work will take two primary directions. First, the physiological accuracy and subject variability of the phantom geometry will be revised. Using medical imaging from swine and human studies, better placement and geometries for the vessels can be determined and used to create relevant subject variability. Second, non-gelatin phantom alternatives will be evaluated so that the shelf-life for the phantom can be improved. Third, this model will be combined with commercial trainers, ex vivo models, and live animal models into a testing pipeline structure to help medical device developers identify what model complexity is required at different stages of product development to ensure the proper model form is being used to quantify certain metrics.

## 5. Conclusions

In conclusion, the tissue phantom model presented addresses a current testing gap for developing central vascular access medical devices. Troubleshooting is essential and having to rely on live, or even ex vivo, animal testing for evaluating a medical device with physiological mimicking fluid flow or pressure levels will slow device development. The tissue phantom developed here is easy to setup, its design can be tuned to mimic human or swine anatomy, and it can allow for normal and hypovolemic vascular access testing. Vascular access can be tracked by ultrasound, with or without doppler, and quality of the needle insertion can be assessed by measuring fluid leakage from the vessel. For these reasons, we believe the developed model will assist with automated central vascular access device development which will accelerate product translation to simplifying critical medical procedures during trauma scenarios.

## Figures and Tables

**Figure 1 bioengineering-09-00319-f001:**
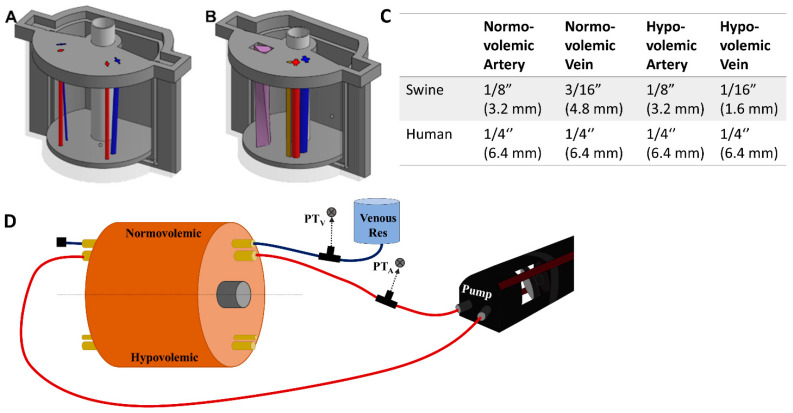
Overview of the phantom design and flow loop setup. (**A**) CAD assembly of the porcine tissue phantom model with arterial (red) and veinous (blue) channel inserts. Left side of the model was for hypovolemic conditions while the right side was normal. (**B**) CAD assembly of the human tissue phantom model with addition of nerve fiber bundle. Left side of the model shows the neurovascular bundle (purple teardrop channel) while the right shows the three individual channels for artery (red), vein (blue) and nerve fiber (brown). (**C**) Summary of the tubing sizes used for normal and hypovolemic conditions for swine and human models. Outer diameters are shown in inches with mm sizes shown in parentheses. (**D**) Flow loop setup for the arterial (red) and venous (blue) channels of the phantom model. Only the normovolemic side is plumbed in the diagram but the same setup would apply for the hypovolemic scenario. Note this setup is not to scale and the phantom model is shown as a large component to the rest of the hardware to highlight features within the phantom.

**Figure 2 bioengineering-09-00319-f002:**
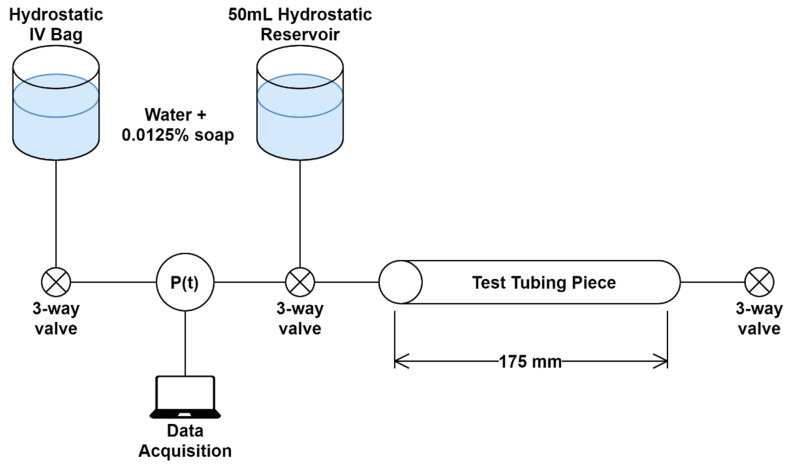
Overview of the tube leak testing. The test tube length was standardized at 175 mm with 3-way valves positioned on both sides to isolate the tubing piece. Two hydrostatic reservoirs were used for the system. One was a pressurized IV bag that could set the starting pressure in the system. The second was for removing air bubbles that may arise. Data was acquired by in-line pressure transducer (P(t)) and LabChart software.

**Figure 3 bioengineering-09-00319-f003:**
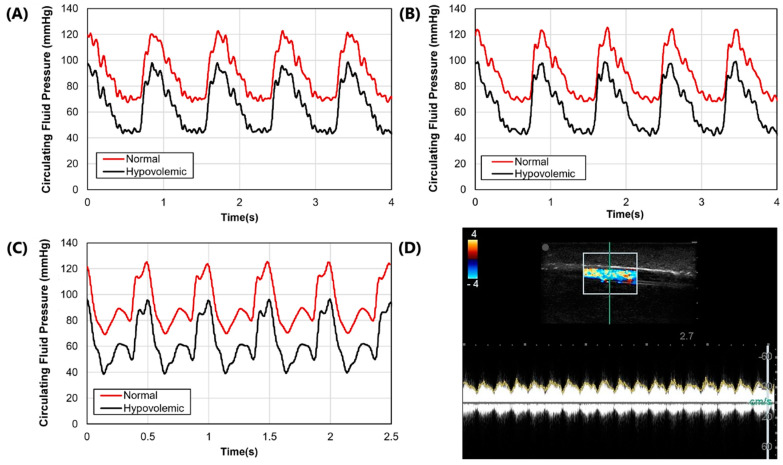
Arterial waveform results for the phantom platform. (**A**–**C**) Three distinct arterial waveforms were generated at both normal and hypovolemic levels. Each plot shows at least 3 waves for both pressure settings. (**D**) Arterial in-plane doppler signal for the tissue phantom after adding flour to the arterial flow.

**Figure 4 bioengineering-09-00319-f004:**
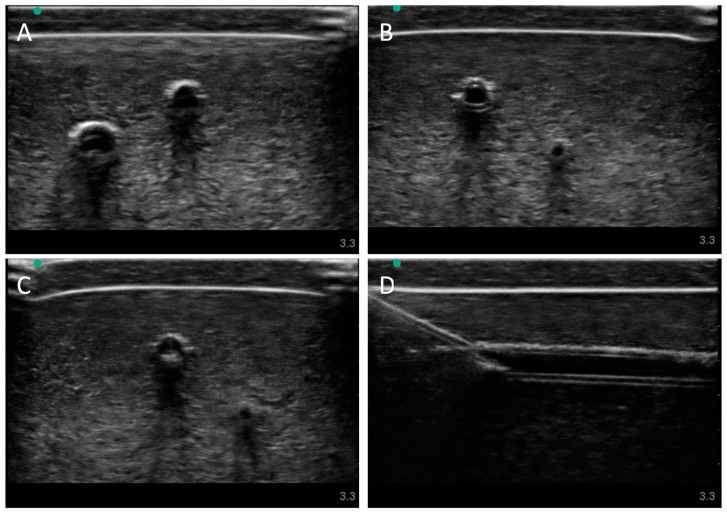
Porcine gelatin phantom ultrasound results. (**A**) Normovolemic OOP view of artery and vein, (**B**) hypovolemic OOP view of artery and vein, (**C**) hypovolemic OOP view with needle insertion into the artery with the needle tip visible close to lower vessel wall, and (**D**) hypovolemic IP view of needle insertion into the vein with needle bevel visible near the lower vessel wall.

**Figure 5 bioengineering-09-00319-f005:**
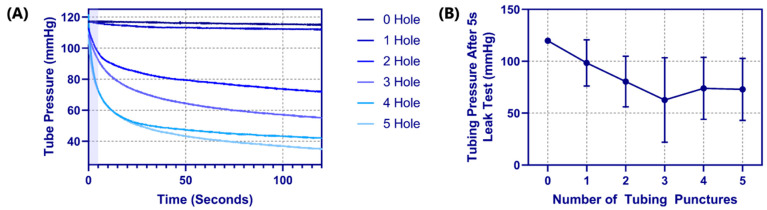
Tubing leak test results for the phantom platform. (**A**) Pressure decay vs. time plots for a single tubing piece with 0 to 5 holes in the tubing, for 120 s. The shaded region indicates the 5 s window that was used for comparing across tubing pieces. (**B**) Results for tubing pressure after 5 s into the leak test are shown for 0 to 5 puncture holes in the tubing pieces. Results are shown as mean (*n* = 4 tubing pieces) with error bars denoting standard deviations.

**Figure 6 bioengineering-09-00319-f006:**
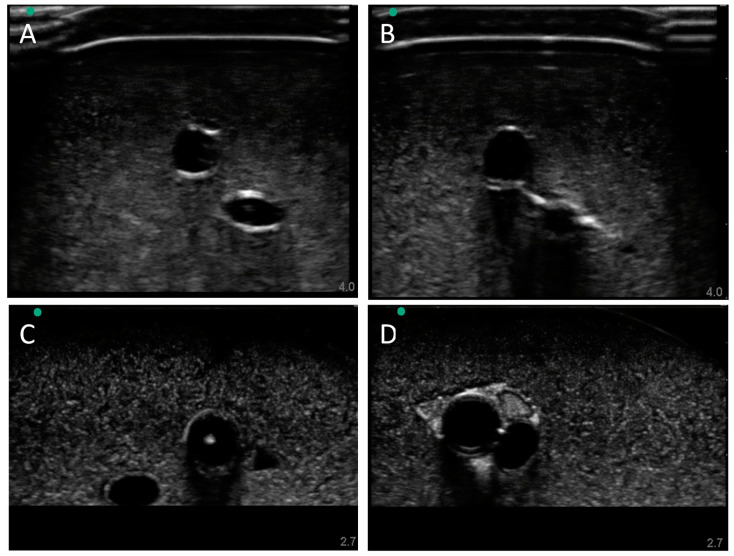
Human gelatin phantom ultrasound results. (**A**) Normovolemic conditions with a skin layer integrated in the model, showing both artery and vein for OOP ultrasound view. (**B**) Hypovolemic pressure conditions with a skin layer showing artery and vein for OOP ultrasound view. (**C**) Artery, vein, and nerve fiber bundle with distinct channels in the human phantom anatomy, no skin layer. OOP view is shown with a needle tip in view in the artery. (**D**) Human neurovascular bundle shown as OOP view.

**Table 1 bioengineering-09-00319-t001:** Summary of normal and hypovolemic circulating fluid pressure readings for arterial and venous flow. Results are shown for three waveform examples and as mean results. For simplicity, arterial flow is described as systolic, diastolic, and mean arterial pressure (MAP), while central venous pressure (CVP) is shown for venous flow. All values are in units of mmHg.

Pressures	Waveform 1		Waveform 2		Waveform 3		Average	
	Normal	Hypo-volemic	Normal	Hypo-volemic	Normal	Hypo-volemic	Normal	Hypo-volemic
Systolic	125	99	129	95	122	98	125.3	97.3
Diastolic	68	42	71	38	68	43	69	41
MAP	90	64	94	63	90	66	91.3	64.3
CVP	15	7	15	7	15	7	15	7

**Table 2 bioengineering-09-00319-t002:** Comparison of the modular tissue phantom platform vs. commercial vascular trainers. Checkmark indicates model can successfully perform. * Depths and distances between vessels vary across the most commercial phantoms but cannot be adjusted. ** Commercial trainers are likely normovolemic compatible but without pressure control, cannot confirm.

Model Features for Evaluating Vascular Access	Modular Tissue Phantom Platform	Commercial Vascular Trainers
Can measure needle bore tip relative to all vessel walls	✓	✓
Angle of needle to surface can be visualized	✓	✓
Needle flashback for successful insertion into vessel	✓	✓
Subject vessel depth variability	✓	*
Subject distance between vessels variability	✓	*
Subject vessel diameter variability	✓	
Normovolemic Compatible	✓	**
Hypovolemic Compatible	✓	
Doppler Compliant	✓	
Measure damage from needle insertion	✓	

## Data Availability

The datasets generated during and/or analyzed during the current study are available from the corresponding author upon reasonable request.

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
