# Peer review of "Development of a Modular Tissue Phantom for Evaluating Vascular Access Devices"

_bioengineering, 2022, doi:10.3390/bioengineering9070319_

Round 1
Reviewer 1 Report
The authors report the development of a human and porcine model for use in training to fully automated central venous catheter placement for battlefield and emergency situations. This model is designed to reproduce collapsed vessels, enable intravascular pressure measurement and vascular ultrasound, and allow posterior evaluation of the degree of vascular damage, a development that is expected to have advantages such as more realistic and practical training effects and rigorous safety evaluation. This paper is very beneficial to the advancement of this field, and I believe that it is a valuable report that is well-deserving of submission to the Journal of Bioengineering.
Major Comments.
1. The authors have created a porcine model, but no mention is made of the need for this model. The authors should discuss the implications of utilizing a porcine model rather than a living pig or a human model. In addition, the authors should indicate the appropriateness of the settings such as vessel diameter and depth from the epidermis in creating the pig model by citing the literature.
2. Regarding the human model, an explanation of its anatomical validity should be added. Also, the results of the pressure waveform study and Doppler echo analysis are not shown in the human model. I wonder if there is any reason for this. The same applies to the evaluation of post-puncture damage, and the authors should provide the equivalent of Figures 3 and 5 for the human model.
Minor Comments.
1. Regarding Figure 1, to improve understanding of this development, it would be better to include a photograph of the actual product rather than just a schematic diagram.
2. Regarding Figure 3, the vertical axis of the graph of the pressure curve is marked as blood pressure, but it should be written as pressure or intra-circuit pressure because it is not the actual blood pressure of a human body, but the pressure in the circuit that mimics the arterial pulse flow. The same point applies to Table 1 as well, and it would be less confusing if it were revised.
Reviewer 2 Report
The authors submitted a research article in which they thoroughly elucidate novel tissue phantom model produced of gelatin and cast in a 3D printed mold with channels for vessels. The aim of the study is clear and deserves to be investigated. The manuscript has a logical structure, composses of well-balanced sub-sections and informative discussion along with conprehensive conclusive part. The figures of the article are informative and legible. I have no major concerns about the study and congratulate the authors on it. However, I would like to put forward several issues to discuss.
1. It remains unclear what benefits of the tissue phantom model when compared with previously developed models are. Please, check and clearly elucidate.
2. Did authors validate the model or not ?
3. Please, add a short comment in the section Discussion about all sorts of ways by which this model can be used in education and skill poling
Reviewer 3 Report
Here, Boice and colleagues developed a customised gelatin tissue phantom with inserts for blood vessel imitation, aimed to advance training of central vascular access, a potentially life-saving procedure which requires a skilled personnel. Among the advantages of the model is a modular design, allowing to adjust blood vessel size and location, and even permitting to mimic neurovascular bundles, fully simulating native human or swine vascular anatomy and syntopy. Generation of a pulsatile, cardiac-mimicking arterial flow allows modelling of different waveforms and multiple clinical scenarios.
Overall, the invention is quite original and the paper nicely describes its clinical relevance. I can recommend it for acceptance.
